# Contraceptive Behaviors in Polish Women Aged 18–35—A Cross-Sectional Study

**DOI:** 10.3390/ijerph16152723

**Published:** 2019-07-30

**Authors:** Magdalena Zgliczynska, Iwona Szymusik, Aleksandra Sierocinska, Armand Bajaka, Martyna Rowniak, Nicole Sochacki-Wojcicka, Miroslaw Wielgos, Katarzyna Kosinska-Kaczynska

**Affiliations:** 1Students’ Research Group at the 1st Department of Obstetrics and Gynecology, Medical University of Warsaw, Starynkiewicz Square 1/3, P.O. Box 02-015 Warsaw, Poland; 21st Department of Obstetrics and Gynecology, Medical University of Warsaw, Starynkiewicz Square 1/3, P.O. Box 02-015 Warsaw, Poland

**Keywords:** contraception, contraception methods, contraceptive counselling, reproductive health, family planning

## Abstract

The aim was to evaluate contraceptive behaviors, and factors affecting them, in the population of Polish-speaking women. A cross-sectional study was performed on 6763 women, current contraceptive users, aged 18 to 35. An anonymous and voluntary questionnaire written in Polish, containing 33 questions, was distributed online from January to February 2017. The Internet and doctors were the most popular sources of information about contraception (82% and 73%, respectively). Upon choosing contraception, women paid the most attention to its efficacy (85%) and its impact on health (59%). The most common methods were combined oral contraceptives (38%) and condoms (24%). In total, 51% had chosen hormonal contraception, of which 68% experienced side effects. The most frequent were decreased libido (39%) and weight gain (22%). Factors associated with the usage of hormonal or non-hormonal contraception were: education, relationship status, parenthood, number of sexual partners, frequency of intercourses, sources of information about contraception, and factors considered most important when choosing a contraceptive method. The choice between short-acting and long-acting reversible contraception was influenced by age, relationship status, parenthood, smoking, sources of information about contraception, and factors considered most important when choosing a contraceptive method. Wide access to contraception, high-quality education, and counselling should become priorities in family planning healthcare.

## 1. Introduction

Contraception aims to prevent unwanted pregnancy. Nowadays, various contraceptive methods are used by the majority of reproductive-age women in formal and informal relationships worldwide [1]. However, there are significant differences across regions and countries. According to the model-based estimates from the United Nations Family Planning Report 2017, contraceptive prevalence in Eastern Europe ranges from 65% in Moldova up to 76% in Czech Republic, with estimated 70% in Poland [2]. Nevertheless, the latest data from Statistics Poland [3] showed that over 61% of women aged 15–50 who declared sexual activity used contraception, and that their numbers were steadily increasing.

Despite the wide availability of contraception and the increase in its use over the past decades, unmet need for family planning, which is defined as unfulfilled willingness to delay parenthood, affects at least one in 10 women of reproductive age in Europe [4]. It is estimated that this problem concerns over half a million women in Poland [1]. Moreover, rates of unintended pregnancies are still very high throughout Europe, reaching up to 43 per 1000 women aged 15–44, per year [5].

Nowadays, there is a wide range of contraceptive methods to choose from [6]. They are classified as hormonal or non-hormonal, short-acting or long-acting, and reversible or irreversible [7,8,9]. Such a wide range of birth control methods enables appropriate adjustment to patient’s expectations, but it does not make the choice easier for either the woman or the doctor. The choice of contraception is difficult for patients, commonly accompanied by feelings of stress and uncertainty [10].

The aim of the study was to assess current contraceptive behaviors in Polish women and to investigate their attitudes toward particular methods. Furthermore, the aim was to assess the level of patient satisfaction with current contraception and contraceptive counselling in Poland. Moreover, the goal was to increase awareness of this important topic, as well as to draw attention to areas in which improvements could be made.

## 2. Materials and Methods

### 2.1. General Information

A survey-based, cross-sectional study was performed in January and February 2017. A questionnaire written in Polish was distributed via the Internet on Facebook and Instagram profiles of Mamaginekolog, a blog aimed primarily at women, currently followed by half a million people. The study was conducted in accordance with the Declaration of Helsinki. The survey was completely voluntarily and anonymously—it did not contain any questions about personal data that would enable the identification of participants, and only the authors of the study had access to the collected data.

### 2.2. Survey Questionnaire

The self-composed questionnaire consisted of 33 questions. The title page contained information about the subject and the objectives of the study, the approximate time needed for its completion, names of the authors and appropriate contact details. The following pages contained 4 questions regarding basic demographic data, 3 questions about the respondent’s obstetric history, and 5 questions about their general medical history, lifestyle, and any addictions. The majority of questions (n = 21) concerned contraceptive behaviors. Respondents were asked which contraceptive method or methods they currently used, if it was their first contraceptive method, and about reasons for potential changes of contraceptives in the past. Study participants also provided information about their level of satisfaction with their current contraceptive method and its side effects. They were presented with questions about reasons for their current choices, and about their sources of information about contraception. Respondents were requested to give details of the medical consultation prior to receiving hormonal contraception. They were also asked about their attitude towards issues such as vasectomy and full reimbursement of contraception.

The inclusion criteria were: women aged 18–35, current contraception use, and being sexually active. All answers were checked for duplicates, and no identical records were found.

### 2.3. Statistical Analysis

Statistical analysis was performed using Microsoft Excel (Microsoft Corporation, Redmond, WA, USA) and STATISTICA 13.3. software (TIBCO Software Inc., Palo Alto, CA, USA). Descriptive data was presented as numbers, percentages, and means with standard deviations. The strength of association between two events was measured using the odds ratio (OR), with statistical significance determined using the chi-squared test. In addition, two multiple logistic backward stepwise regression models were developed—to investigate factors associated with the use of hormonal contraception, and to reveal factors associated with the use of long-acting reversible contraceptives (LARCs). Results of those analyses were presented as adjusted odds ratios (aOR), and 95% confidence intervals (95%CI). Statistical significance was determined for *p*-values (*p*) < 0.05.

## 3. Results

### 3.1. Study Population

The survey was completed by 7085 women, of which 95% (n = 6763) filled it out correctly. The rest provided contradictory or mutually exclusive information, or did not meet all the inclusion criteria. In total, 6763 questionnaires were analyzed. Characteristics of the study group are presented in Table 1.

### 3.2. Sources of Information about Contraception

In a multiple-choice question regarding sources of information about contraception, the Internet and physicians were the most frequent choices (82%, n = 5579, and 73%, n = 4914, respectively). Women were least likely to learn about contraception in school (8%, n = 562) and from parents (5%, n = 367). Respondents also learned about contraception from books and magazines (33%, n = 2247), friends (30%, n = 2050), and drug leaflets (29%, n = 1979).

### 3.3. Choice of Contraception

The question about currently used contraception was also multiple choice. The most popular methods were condoms and combined oral contraceptives (COCs) (Table 2). 51% of women (n = 3426) had chosen hormonal contraception, both short-acting (COCs, vaginal rings, transdermal patches, progestogen-only pills) and long-acting (hormonal IUDs, implants, or medroxyprogesterone injection). Including users of copper IUDs (2%), LARCs were chosen by 5% of participants (n = 317).

In total, 33% of respondents (n = 2234) used more than one method at the same time. The most common combinations were those with condoms: condom and oral contraception (COCs or progestogen-only pills) (28% of multiple contraceptive method users, n = 621 out of 2234), condom and withdrawal (18%, n = 403 out of 2234) and condoms and natural methods (14%, n = 315 out of 2234).

38% of women (n = 2580) were using the type of contraception they had chosen originally. The rest (62%, n = 4183) had already changed the method of contraception because of multiple reasons, among them: side effects (40%, n = 1670 out of 4183), troublesome usage (20%, n = 829 out of 4183), unsatisfactory effectiveness (12%, n = 491 out of 4183), suggestions of partners or friends (5%, n = 196 out of 4183).

### 3.4. Factors Influencing the Choice of Contraception

Participants were asked about factors which influenced their selection of a contraceptive method (multiple-choice question). The most important factors were efficacy (85%, n = 5767), possible impact on health (59%, n = 3994), and comfort of use (44%, n = 2964), while the least selected answers were price (13%, n = 850) and ideological issues (5%, n = 308).

Additionally, around half out of 3426 women, who had chosen hormonal contraception, wanted to obtain additional positive effects beside contraception, such as improvement of skin condition or decrease in menstrual blood loss (52%, n = 1772).

Two separated multiple logistic regression analyses were used to investigate factors related to the use of hormonal contraception (Table 3) and LARCs (both hormonal and non-hormonal) (Table 4). In both cases, the following factors were taken into account when performing univariate analyses: age, education, place of residence, income, relationship status, offspring, number of sexual partners during lifetime, frequency of intercourse, physical activity, smoking, sources of information about contraception (doctors, the Internet, drug leaflets, books and magazines, friends, school, parents), factors considered most important in choice of contraceptive method (efficacy, impact on health, price, comfort of use, ideological issues). All variables which were statistically significantly related (*p* < 0.05) to the use of a particular method in the univariate analyses were included in the final multifactorial model using a backward stepwise approach.

In the first analysis (use of hormonal contraception= yes), all factors except “parents as a source of information about contraception” were statistically significant in univariate analyses and were therefore included in the final multifactorial model.

In the second analysis (use of LARCs = yes) the following factors were statistically significant in univariate analyses: age, relationship status, offspring, number of sexual partners during lifetime, smoking, sources of information about contraception (doctors, books and magazines, school), factors considered most important in choice of contraceptive method (efficacy, impact on health, price, comfort of use, ideological issues), and were therefore included in the final multifactorial model.

### 3.5. Satisfaction with Current Method of Contraception

In the question regarding satisfaction with the current contraceptive method, participants could choose one of the following options: “I am satisfied and I do not want to change the currently used method”, “I am satisfied, but I would like to try another method” and “I am dissatisfied with the currently used method”. Of the results, 90% of women (n = 6099) were satisfied with currently used contraception, however 28% of them (n = 1738) would have liked to try another method. In total, 10% of respondents (n = 665) were not satisfied with presently used birth control. Table 5 shows chances of achieving satisfaction with current contraceptive method in comparison with condoms only. To achieve objective results, only respondents who declared the use of a single method of contraception at the time of completing the survey were included in the analysis.

### 3.6. Side Effects of Hormonal Contraception

In a multiple-choice question about side effects, two out of three women (68%, n = 2340) using hormonal contraception reported at least one. Decreased libido was the most frequent undesirable effect (39%, n = 1336). Other frequent side effects included weight gain (22%, n = 769), mood disorders (21%, n = 731), headaches (17%, n = 585), abnormal bleeding (15%, n = 527), and tenderness of breasts (14%, n = 478). The least reported symptoms were: nausea and vomiting (5%, n = 176), swelling (5%, n = 168), and thrombosis (< 1%, n = 20). Table 6 shows the risks of most common adverse reactions for selected contraceptives. COCs, as the most frequently used hormonal contraception in the study group, were used as reference.

### 3.7. Contraceptive Counselling

In total, 76% of patients (n = 5161) had already known which contraceptive method they wanted to use prior to seeing a doctor, but almost a quarter (24%, n = 1602) preferred to decide based on the physician’s suggestion.

For most women, who made their own decision regarding contraceptives, the doctor concurred with their choice (92%, n = 4747). Some of them altered their choice upon the physician’s recommendation (8%, n = 414).

Less than half of the participants had ever consulted a doctor for advice regarding contraception (41%, n = 2773). Within this group, 67% (n = 1864) were satisfied with the advice given. Almost one-third of women (33%, n = 909) who asked for advice found the answer unsatisfactory. In only 8% (n = 535) of cases the conversation was initiated by the doctor.

There were 840 out of 3426 women (25%), who had chosen hormonal contraception and did not undergo the required examination (medical interview and blood pressure measurement). It means that, in this particular group, hormonal contraception was implemented only upon the request of the patient.

### 3.8. Other Raised Issues

Emergency contraception was used by 24% of the respondents (n = 1643), and only once in most cases (81%, n = 1327).

Of the respondents, 6% (n = 435) conceived despite the use of contraception. Unfortunately, it was impossible to determine which method had been used at the time or whether it had been used properly.

Of women, 17% (n = 1177) would like their partner to undergo vasectomy, while 27% (n = 1847) do not have an opinion on this subject.

Of the respondents, 62% (n = 4226) would like total reimbursement of contraception, whereas 18% (n = 1233) were against it. The remaining participants (19%, n = 1304) did not have an opinion on the matter.

## 4. Discussion

There is no doubt that worldwide trends in contraception have undergone huge changes over the past few decades [11,12,13]. Despite multiple choices, COCs remain one of the most commonly used contraceptive methods in big European countries (France, Germany, Italy, Spain and the United Kingdom), and are substantially more popular than LARCs [14]. However, based on the 2009 report from Statistics Poland [3,15], barrier methods, especially condoms, were the most common choice among Polish women. In this study, they remained the most popular form of birth control, including combinations with other methods. Moreover, about one-third of respondents declared using highly unreliable methods, such as withdrawal or observational methods. The reason for this situation in Poland is not clear. Therefore, this research also focused on the assessment of factors influencing the choice of contraception among Polish women. However, it should be noted that contraceptive patterns vary greatly depending on the country and the culture [16]. The scope of this study did not include a comparison between Polish-speaking women and other groups, but such differences were demonstrated in larger reports [1,15]. A study on the Polish population, conducted by Colleran and Mace, suggested that sociocultural influences on contraceptive behaviors seem to be even stronger than the characteristics of an individual [17] while Nowosielski, et al. [18] emphasized also the importance of spirituality, self-esteem and sexual self-schema for the contraceptive methods decision-making.

Obviously, patients’ priorities also play a crucial role in the decision making. The women who participated in this study paid special attention to efficacy, possible health impacts and comfort of use, which translated, to some extent, into a choice between hormonal and non-hormonal contraception, as well as short-acting methods and LARCs. Return to infertility, sexually transmitted infections prevention, menstrual changes, reputation, and many more, also have been proven to be of significance [19].

Sexual education and sources of information about contraception are also very important factors that influence contraceptive behavior [20,21,22]. A recent online survey conducted by Warzecha, et al. [23], showed that the level of education about reproductive health in Poland is insufficient. Unfortunately, this problem seems to affect many countries around the world [24,25,26]. It is worth pointing out that, in this study, only 8% of participants learned about contraception in school. Additionally, it is alarming that women who learned about contraception in school were less likely to choose reliable methods. In contrast to the above, women who were educated by doctors chose more reliable contraceptive methods. This finding may also indicate poor quality of family planning education in Poland, besides its incredibly low prevalence. This is significant in light of the fact that the vast majority of Polish women undergo their sexual initiation during teenage years—according to Durda-Masny, et al. [27] in women born in 1991–1995 the age of initiation was about 16 years, while Olszewski, et al. [28] showed that the most commonly used contraception during sexual initiation are condoms (68%), and that most people (67%) do not alter the original choice of contraception in their future sexual encounters. Therefore, it is crucial to provide accessible contraceptive counselling with an individualized approach and inclusion of the patient in the decision-making process. Women want the contraception provider to participate in the selection process [29], and many of them, upon receiving proper counselling, are ready to change their current contraceptive [30].

There is a number of studies focused on factors that limit the use of modern contraception, and the following are indicated: concerns regarding side effects, fear of infertility [31], financial issues, medical and legal restrictions [32], religious and personal beliefs [16], myths and misinformation [33], and many others.

This study also shows that a significant percentage of women using hormonal contraceptives report side effects, and that these are the most common cause of its discontinuation. Decreased libido was the most frequently reported adverse effect (39%), and similar observations were also made by other authors [34,35]. However, there are indications that this is a consequence of a complex combination of diverse factors: biological, psychological, and social [36,37,38]. A similar observation can be made concerning mood disorders, reported by 21% of hormonal contraception users [39]. The second most frequent undesirable effect reported by women was weight gain (22%), although the influence of combined contraception on body weight has not been unambiguously confirmed in studies published to date [40,41].

The majority of the above side effects are mild and do not endanger women’s health, nonetheless, with such high prevalence (68% of hormonal contraception users in the study), emphasis should be placed on informing patients about the possibility of these side effects occurring, identification of potential predisposition risk factors, and creation of algorithms for individualization of contraceptive method selection.

However, hormonal contraceptives exert a pleiotropic effect on the body. Many users are aware of this—in the study around half of women, who have chosen hormonal methods, wanted to obtain additional positive effects beside contraception. It is worth noting some of the more common ones: reduction of ovarian, endometrial and colorectal cancer risk [42], improvement of skin condition [43,44], positive effect on bone metabolism in certain groups of women [45], decrease in menstrual blood loss [46], and many others. In various hormonal methods, all of the above effects are present with varying frequency and strength. Therefore, a discussion regarding the patient’s expectations could help her choose the method that will be both satisfactory and increase her quality of life [47,48,49].

Emergency contraception was used by a quarter of the study’s population. With such a high percentage of unwanted pregnancies worldwide, estimated at 43 per 1000 women in Europe, this is a topic that requires special attention [5]. There are practically no medical contraindications for its use [50,51], however, reluctance to prescribe and to sell it in Poland are frequent phenomena [52]. Increased access to emergency contraception, together with the popularization of more reliable contraceptive methods, are of special importance, as this could contribute to the reduction of unwanted pregnancies in the future [53]. Moreover, widespread education about its mode of action could also contribute to its wider use in certain groups of women [54,55].

The advantages of this study include a large sample size and a wealth of data. It is also one of the most up to date studies on contraception in Poland. The most recent large study, performed specifically on the Polish population, was published about 10 years ago [3]. Unfortunately, there is a lot of missing data on contraceptive behavior in Poland, including in many international reports such as The Reproductive Health Report. Moreover, unlike other studies, the presented research also focuses on factors influencing the choice between types of contraceptives (hormonal vs. non-hormonal, short-acting vs. LARCs), and on the functioning of counselling in Poland from the patient’s point of view. Research on contraception is a particularly important topic for the Polish population bearing in mind that Poland is a country characterized by a rigorous approach to family planning–impossibility of obtaining emergency contraception without prescription, restrictive access to induced abortion, and many others.

There are several limitations of this study that should be noted. The study group included Polish-speaking respondents only, therefore sexual and contraceptive behaviors were most likely influenced by the specific cultural context, and the following conclusions may not apply to every country [16,56]. Furthermore, the issue of family planning applies to both men and women, therefore aiming this research only at women presents the topic from a narrowed perspective. Moreover, the data presented in this article derives from a self-composed questionnaire, which could be the cause of an inherent bias in the study. It was distributed online, therefore the question of reliability of the results is a valid concern. However, the Internet was chosen to administer this survey due to its ubiquitous nature. Despite its obvious limitations as a research tool, it allows to reach a much larger and more diverse group of people from all over the country. What is more, the survey was generally aimed at young women, who are more likely to use the Internet. Furthermore, the anonymity of an online questionnaire may reduce the risk of false answers, especially in personal questions [57,58].

Based on the results of the above presented study, suggestions for improvements of the current situation in Poland may be proposed. Sexual education is an area where the largest impact could be made. Due to the widespread use of the Internet as a source of information, websites containing reliable information on contraception should be established and promoted. This solution has already been successfully implemented in many other countries, however, at present no government-supported Polish website on this subject exists on the Internet. However, this solution is not as simple to implement as it may seem. A recently published interesting study by Byker, et al. [59] has shown that even a large social media campaign promoting the use of LARC did not have a detectable impact on LARC insertions in the following months. Sexual education in schools was also indicated to be an area in great need of improvement. According to Polish law, family planning classes are optional for students and the total time allocated for this subject is only fourteen hours. Therefore, it is crucial that the teaching is of the highest quality, and the students learn as much as they can from it. Another issue in need of attention is the availability of contraception and counselling. The task sharing strategy, recommended by the World Health Organization among others, involves allowing other healthcare workers, such as nurses or midwives, to provide contraceptives, which could greatly improve access to contraception. Other solutions include making emergency contraception available without prescription and expanding the indications for contraception reimbursement, which at present are very restrictive.

## 5. Conclusions

It is essential to focus on the implementation of high-quality sexual education as well as on the popularization of highly reliable sources of information about contraception. High-quality, yet easily accessible contraceptive counselling is needed, so that every woman could take advantage of patient-centered family planning healthcare.

## Figures and Tables

**Table 1 ijerph-16-02723-t001:** Characteristics of the study population.

Variable	% (n)
**Age [years old]**	
26.4 (average)	standard deviation 3.9
18–24	34% (n = 2268)
25–29	44% (n = 2955)
30–35	23% (n = 1540)
**Education**	
Primary	<1% (n = 43)
Secondary	29% (n = 1950)
Vocational	1% (n = 75)
Higher	69% (n = 4695)
**Place of residence**	
Rural and city < 5000 inhabitants	20% (n = 1324)
City 5000–200,000 inhabitants	35% (n = 2368)
City 200,000–1,000,000 inhabitants	30% (n = 2000)
City > 1,000,000	16% (n = 1071)
**Estimated income/one family member per month**	
<1000 PLN (~0–230 EUR)	17% (n = 1167)
>1000–2500 PLN (~230–580 EUR)	53% (n = 3596)
>2500–3500 PLN (~580–815 EUR)	16% (n = 1075)
>3500 PLN (>815 EUR)	14% (n = 925)
**Relationship status**	
Married	49% (n = 3312)
Informal relationship	46% (n = 3119)
Single	5% (n = 332)
**Offspring**	
No children	60% (n = 4080)
One child	27% (n = 1806)
Two or more children	13% (n = 877)
**Physical activity**	
Few times a week	18% (n = 1200)
Once a week	13% (n = 900)
Occasionally	55% (n = 3709)
Not at all	14% (n = 954)
**Smoking**	
Yes, regularly	10% (n = 703)
Yes, occasionally	9% (n = 635)
Not at all	80% (n = 5425)
**Chronic diseases**	
Thyroid diseases	13% (n = 867)
Asthma	2% (n = 143)
Diabetes or prediabetes	1% (n = 86)
Hypertension	<1% (n = 51)

**Table 2 ijerph-16-02723-t002:** Contraceptive methods used by the study population.

Contraception Method	
Condoms in total *	50% (n = 3375)
COCs **	38% (n = 2586)
Condoms only	24% (n = 1642)
Withdrawal	17% (n = 1171)
Natural family planning	13% (n = 897)
Vaginal ring	4% (n = 263)
Transdermal patch	3% (n = 202)
Progestogen-only pills	2% (n = 163)
Hormonal IUD ***	2% (n = 142)
Copper IUD	2% (n = 105)
Implant	<1% (n = 54)
Medroxyprogesterone injection	<1% (n = 16)
Female condom	<1% (n = 13)
Vasectomy	<1% (n = 3)

* combined with other methods; ** combined oral contraceptives; *** intrauterine device.

**Table 3 ijerph-16-02723-t003:** Results of the multiple logistic regression of factors associated with the use of hormonal contraception.

**Factor**	**aOR (95% CI) ***	***p*-Value**
**Education**		
Primary/Secondary/Vocational	1.18 (1.04; 1.33)	0.009
Higher	1.00 (reference)	-
**Relationship status**		
Single	1.00 (reference)	-
Informal relationship	0.56 (0.42; 0.76)	<0.001
Married	0.32 (0.24; 0.44)	<0.001
**Offspring**		
No	1.00 (reference)	-
Yes	0.60 (0.52; 0.68)	<0.001
**Number of sexual partners**	1.02 (1.00; 1.04)	0.015
**Frequency of intercourses**	1.12 (1.06; 1.19)	<0.001
**Sources of information**		
*Internet*		
No	1.00 (reference)	-
Yes	0.66 (0.57; 0.76)	<0.001
*Doctor*		
No	1.00 (reference)	-
Yes	2.87 (2.52; 3.27)	<0.001
*Books and magazines*		
No	1.00 (reference)	-
Yes	0.66 (0.59; 0.75)	<0.001
*Drug leaflets*		
No	1.00 (ref.)	-
Yes	1.30 (1.15; 1.47)	<0.001
*Friends*		
No	1.00 (reference)	-
Yes	0.77 (0.68; 0.86)	<0.001
*School*		
No	1.00 (reference)	-
Yes	0.55 (0.45; 0.68)	<0.001
**Factors considered most important in the choice of method**		
*Efficacy*		-
No	1.00 (reference)	
Yes	2.12 (1.79; 2.52)	<0.001
*Impact on health*		
No	1.00 (reference)	-
Yes	0.33 (0.29; 0.37)	<0.001
*Price*		
No	1.00 (reference)	-
Yes	1.34 (1.13; 1.59)	0.001
*Ideological issues*		
No	1.00 (reference)	-
Yes	0.19 (0.13; 0.28)	<0.001

* adjusted odds ratio (95% confidence interval).

**Table 4 ijerph-16-02723-t004:** Results of the multiple logistic regression of factors associated with the use of long-acting reversible contraceptives (LARCs).

Factor	aOR (95% CI) *	*p*-Value
**Age**	1.07 (1.03; 1.11)	<0.001
**Relationship status**		
Single	1.00 (reference)	-
Informal relationship	0.49 (0.29; 0.84)	0.01
Married	0.38 (0.22; 0.67)	0.001
**Offspring**		
No	1.00 (reference)	-
Yes	4.69 (3.43; 6.41)	<0.001
**Smoking**		
No	1.00 (reference)	-
Occasionally	1.29 (0.86; 1.93)	0.224
Yes	1.75 (1.26; 2.45)	0.001
**Sources of information**		
*Doctor*		
No	1.00 (reference)	-
Yes	1.39 (1.02; 1.88)	0.035
*Books and magazines*		
No	1.00 (reference)	-
Yes	0.73 (0.56; 0.96)	0.024
**Factors considered most important in the choice of method**		
*Efficacy*		
No	1.00 (reference)	
Yes	1.93 (1.28; 2.90)	0.002
*Comfort of use*		
No	1.00 (reference)	-
Yes	2.63 (2.06; 3.36)	<0.001
*Price*		
No	1.00 (reference)	-
Yes	0.45 (0.27; 0.74)	0.002
*Ideological issues*		
No	1.00 (reference)	-
Yes	0.21 (0.05; 0.87)	0.032

* adjusted odds ratio (95% confidence interval).

**Table 5 ijerph-16-02723-t005:** Chances of achieving satisfaction with current contraceptive method with condoms only as a reference.

	Satisfaction with Applied Method					
Method	Yes		No			
	n	%	n	%	OR (95%CI) *	*p*-Value
Condoms only (n = 1642)	1401	85%	241	15%	1.00 (reference)	-
Withdrawal only (n = 261)	232	89%	29	11%	1.38 (0.91–2.07)	0.125
Natural family planning only (268)	260	97%	8	3%	5.59 (2.73–11.45)	<0.001
COCs ** only (n = 1807)	1679	93%	128	7%	2.26 (1.80–2.83)	<0.001
Vaginal ring only (n = 224)	215	96%	9	4%	4.11 (2.08–8.12)	<0.001
Transdermal patch only (n = 168)	149	89%	19	11%	1.35 (0.82–2.22)	0.2359
Progesterone-only pills only (n = 134)	110	82%	24	18%	0.79 (0.50–1.25)	0.1951
Hormonal IUD *** only (n = 131)	129	98%	2	2%	11.10 (2.73–45.14)	<0.001

* odds ratio (95% confidence interval); ** combined oral contraceptives; *** intrauterine device.

**Table 6 ijerph-16-02723-t006:** Risk of decreased libido, weight gain and mood disorders for selected contraceptives in comparison with combined oral contraceptives (COCs) as reference.

Side Effects Methods	Yes		No			
	n	%	n	%	OR (95%CI) *	*p*-Value
**Decreased Libido**						
COCs ** (n = 2586)	941	36%	1645	64%	1.00 (reference)	-
Vaginal ring (n = 263)	103	39%	160	61%	1.13 (0.87–1.46)	0.374
Transdermal patch (n = 202)	72	36%	130	64%	0.97 (0.72–1.31)	0.832
Progestogen-only pills (n = 163)	56	34%	107	66%	0.91 (0.66–1.28)	0.601
Hormonal IUD *** (n = 142)	23	16%	119	84%	0.34 (0.21–0.53)	<0.001
**Weight Gain**						
COCs (n = 2586)	617	24%	1969	76%	1.00 (reference)	-
Vaginal ring (n = 263)	46	17%	217	83%	0.68 (0.49–0.94)	0.020
Transdermal patch (n = 202)	51	25%	151	75%	1.08 (0.77–1.50)	0.656
Progestogen-only pills (n = 163)	34	21%	129	79%	0.84 (0.57–1.24)	0.382
Hormonal IUD (n = 142)	10	7%	132	93%	0.24 (0.13–0.46)	<0.001
**Mood Disorders**						
COCs (n = 2586)	611	24%	1975	76%	1.00 (reference)	-
Vaginal ring (n = 263)	34	13%	229	87%	0.48 (0.33–0.70)	<0.001
Transdermal patch (n = 202)	29	14%	173	86%	0.54 (0.36–0.81)	0.003
Progestogen-only pills (n = 163)	27	17%	136	83%	0.64 (0.42–0.98)	0.038
Hormonal IUD (n = 142)	12	8%	130	92%	0.30 (0.16–0.54)	<0.001

* odds ratio (95% confidence interval); ** combined oral contraceptives; *** intrauterine device.

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
