# Peer review of "Contraceptive Behaviors in Polish Women Aged 18–35—A Cross-Sectional Study"

_ijerph, 2019, doi:10.3390/ijerph16152723_

Round 1
Reviewer 1 Report
General: Interesting topic, well presented. Flow of language: acceptable
Title: suitable
Text structure and content: both are adequate
The reference list: most authors with important research in refered area are included.
Author Response
Dear Reviewer,
Thank you very much for your time and effort in reviewing this article.
Yours faithfully,
Iwona Szymusik
Reviewer 2 Report
Topic has been described a lot of times before and even though is not novel may be very new for polish women.
English language is adequate.
Material and methods are well described.
Results are well presented.
Sample is large and gives strength to the study.
Discussion needs corrections regarding extensive correlation and comparison with other similar studies in Poland and abroad.
Author Response
Dear Reviewer,
Thank you very much for your time and effort in reviewing this article.
As suggested, we have included comparisons to similar studies carried out in Poland and Europe in the discussion (among others new references: 17, 18, 23, 52). Moreover, we have applied some linguistic corrections.
Yours faithfully,
Iwona Szymusik
Reviewer 3 Report
Contraceptive behaviors in Polish women aged 18-35 – a cross-sectional study
Reviewers comment
Contraceptive behaviors in Polish women aged 18-35 is an important issue to address. The research study may help to develop policies to address health of women and the contraceptive needs of the population– an often-neglected issue. Some comments to help authors improve their paper.
It was not clear why this study is being conducted? Is it only to know about the contraceptive behaviour of the polish women for academic purpose or the authors wanted to raise awareness on this important issue. If so they should include implications of the research. Include solutions based on the research findings on how to improve information, counselling and services to improve access and quality of contraceptive services in the context studied.
It was not clear as to what NEW knowledge this paper is adding to the current prevailing situation
The study/questionnaire is only for women- thus reinforcing that FP is only a woman’ s issue. It is important to have men and women’s perspective. (It is not sure who were the actual respondents who filled up the forms). Consider including this caveat in limitations.
Use of hormonal contraceptive- The author tends to use hormonal contraception as different from LARCs. It was not clear which contraceptives are included under hormonal contraceptives. COCs are hormonal contraception and so are some of the LARCs. The paper should be reviewed to make clear which contraceptive is being referred to as HC.
It was not clear how satisfaction level is assessed. Please include the matrix to assess satisfaction
Please check headings of tables 6. Though it states side effects are in comparison to COCs, the table does not show this comparison.
The selection process of the group (internet savy users, followers of a particular blog ) could result in only educated women becoming part of the group ( as seen in the Data) . Include reasoning behind choosing internet only as the mode to conduct this study
As Internet and doctors were the most popular sources of information about contraception, some recommendation on need to scrutinize and check available information on contraception on internet and ways to strengthen counselling
Statistical analysts, discussion, results, conclusion section needs to further analyses, strengthened and linked.
Implications for action be added.
It would be appropriate to make recommendations to train other health workers, for example nurses, midwives, community health workers, who could provide information, counselling and some methods, such as pills and condoms (task sharing is a common practice prevalent in many countries).
Some references need an update.
Please check WHO website for latest on contraceptives including classification. https://www.who.int/reproductivehealth/publications/family_planning/policybriefs/en/
I would suggest the authors consider these comments to update the document accordingly.
Thank you,
Author Response
Dear Reviewer,
Thank you very much for your time and effort to review our manuscript. We are grateful for all your critical suggestions and valuable advice. We have endeavored to comply with your comments as closely as possible. In order to respond to your review fully, we will reply to each of your comments separately:
It was not clear why this study is being conducted? Is it only to know about the contraceptive behaviour of the polish women for academic purpose or the authors wanted to raise awareness on this important issue. If so they should include implications of the research.
Thank you very much for paying our attention to the above fact. Indeed, apart from acquiring knowledge about the contraceptive behaviors and family planning in Poland, our additional goal was to raise awareness of this meaningful topic among potential readers. Therefore, we have included this statement in the text (under the aim of the study).
Include solutions based on the research findings on how to improve information, counselling and services to improve access and quality of contraceptive services in the context studied.
We have added a paragraph in the discussion devoted to possible future actions focused on improving family planning services in Poland. Thank you.
It was not clear as to what NEW knowledge this paper is adding to the current prevailing situation
This paragraph has been extended in the manuscript as suggested. The last available large study performed specifically on the Polish population was performed about 10 years ago (Statistics Poland 2009). Unfortunately, there is a lot of missing data in many international reports, including The Reproductive Health Report, on contraceptive behaviors in Poland. Moreover, our study also focuses on other aspects: factors influencing the choice between groups of contraceptives (hormonal vs non-hormonal, short-acting vs LARC) and on the functioning of counselling in Poland from the patient's point of view. Research on contraception is a particularly important topic for the Polish population in light of the fact that Poland is a country characterized by a rigorous approach to family planning - lack of possibility of obtaining emergency contraception without prescription, restrictive access to induced abortion, and many others. Therefore, we believe that the results of this publication will draw attention to aspects that could be further improved.
The study/questionnaire is only for women- thus reinforcing that FP is only a woman’ s issue. It is important to have men and women’s perspective. (It is not sure who were the actual respondents who filled up the forms). Consider including this caveat in limitations.
Indeed, we have forgotten about that particular limitation. We have added this important observation to that section. We are grateful for pointing that out.
Use of hormonal contraceptive - The author tends to use hormonal contraception as different from LARCs. It was not clear which contraceptives are included under hormonal contraceptives. COCs are hormonal contraception and so are some of the LARCs. The paper should be reviewed to make clear which contraceptive is being referred to as HC.
Thank you very much for noticing those inaccuracies. Indeed, this was not clearly stated in the text, but we purposely used those terms separately, bearing in mind that some of LARC are also hormonal contraceptives. Our aim was to conduct independent analyzes for groups: long-acting contraceptives and short-acting groups as well as for groups: hormonal and non-hormonal contraceptives. We decided to perform statistical analysis for LARC separately, because those very effective and well-known methods are still underused in Poland (in our study only 5% of participants). So the purpose of such analyzes was to asses if there are any special features that distinguish Polish women who choose LARC. Referring to this comment, we have added relevant notes to clarify which methods of contraception have been included in particular analyses.
It was not clear how satisfaction level is assessed. Please include the matrix to assess satisfaction.
We have added relevant information in the manuscript text. Thank you.
Please check headings of tables 6. Though it states side effects are in comparison to COCs, the table does not show this comparison.
Indeed, we have chosen COCs, the most frequently used hormonal contraception in our study, as a reference for the other values. Therefore, calculated odds ratios for subsequent methods refer to 1.00 as a reference value adopted for COCs. In order to increase the readability of results presented in the table, we improved the headings and added the "reference" note in appropriate places. Thank you for pointing this out.
The selection process of the group (internet savy users, followers of a particular blog ) could result in only educated women becoming part of the group ( as seen in the Data). Include reasoning behind choosing internet only as the mode to conduct this study
We included reasoning behind choosing the Internet as a mode to conduct this study and added a note in the limitations section about the possibility of a higher level of education among the surveyed group than in the general population. Thank you.
As Internet and doctors were the most popular sources of information about contraception, some recommendation on need to scrutinize and check available information on contraception on internet and ways to strengthen counselling
Statistical analysts, discussion, results, conclusion section needs to further analyses, strengthened and linked.
Implications for action be added.
It would be appropriate to make recommendations to train other health workers, for example nurses, midwives, community health workers, who could provide information, counselling and some methods, such as pills and condoms (task sharing is a common practice prevalent in many countries).
Some references need an update.
In the new version of the manuscript, we included all of the corrections proposed above. Thank you for these valuable suggestions. We have added some up-to-date literature to the manuscript – references numbers: 17 (2015), 18 (2019), 23 (2019), 52 (2016), 59 (2019).
Please check WHO website for latest on contraceptives including classification. https://www.who.int/reproductivehealth/publications/family_planning/policybriefs/en/
We have also applied some linguistic corrections.
Moreover, we also have applied linguistic corrections. Once again, we would like to extend our sincere gratitude for this detailed review and for drawing our attention to many important issues in the manuscript.
Yours faithfully,
Iwona Szymusik